# Ozone pollution may limit the benefits of irrigation to wheat productivity in India

Gabriella Everett[1], Øivind Hodnebrog[2], Madhoolika Agrawal[3], Durgesh Singh Yadav[4], Connie O'Neill[5], Chubamenla Jamir[6], Jo Cook[1,5], Pritha Pande[1], Sam Bland[5], Lisa Emberson[1*]

[1]Department of Environment and Geography, University of York, York, UK- YO10 5DD.
[2]CICERO Center for International Climate Research – Oslo, 0318, Oslo, Norway
[3]Department of Botany, Institute of Science, Banaras Hindu University, Varanasi 221005, India.
[4]Department of Botany, Government Raza P.G. College, Rampur, U.P. 244901, India.
[5]Stockholm Environment Institute, University of York, York, UK- YO10 5DD.
[6]Climate Studies and Knowledge Solutions Centre, Kohima, Nagaland, India.

*Correspondence to*: Lisa Emberson (l.emberson@york.ac.uk)

**Abstract.** Ground level ozone ($O_3$) pollution, heat and water stress are recognised as key abiotic stresses which threaten the ability of wheat yields to meet the growing demand for food production in India. The magnitude and interplay of $O_3$ and water-stress effects are tightly coupled via stomatal conductance and the transpiration pathway. Existing modelling methods that assess stress response as a function of $O_3$- and water vapour-stomatal flux are applied to assess $O_3$'s role in limiting productivity afforded by irrigation. We investigate the effect of these stresses on grain yield of older (HUW-234) vs recently released (HD-3118) Indian wheat cultivars under recent past     and future climates and $O_3$ precursor emission profiles (using RCP4.5 and RCP8.5 scenarios). Water-stress in rainfed conditions was modelled to analyse the trade-off between $O_3$-induced vs. water-stress-induced yield loss to quantify the extent to which water-stress mitigates $O_3$ stress via reduced stomatal conductance. Under rainfed conditions for the years 1996-2005, the mean water-stress-induced and $O_3$-induced yield loss for HUW-234 was 13.3% and 0.6% respectively. The latter was a significant decrease from the mean $O_3$-induced yield loss of 10.6% modelled under irrigated conditions (i.e. no water stress). Similarly, under RCP4.5 and RCP8.5 scenarios for the mid-century, water-stress induced yield losses under rainfed conditions were 10.1% and 20.0%, while mean $O_3$-induced yield losses were only 1.0% and 0.1% respectively. Under irrigation, $O_3$-induced yield losses increased to 18.5% and 13.7%, suggesting that $O_3$ stress will negate the beneficial effects of irrigation. The cultivar HD-3118 suffered on average 0.2% greater $O_3$ relative yield loss ($O_3$RYL) than HUW-234 across all scenarios. The $O_3$RYL increased with climate change under the RCP4.5 scenario by 7.9% and RCP8.5 by 3.0% compared to the recent past     climate. Together these findings suggest that $O_3$ may continue to substantially limit the productivity benefits of the use of modern cultivars bred for high gas exchange grown under irrigated conditions in India.

## 1 Introduction

Wheat is a vital crop for India's economy and food security; India is the second largest wheat producer in the world and most of its population gains >50% of their calorific intake from this staple grain (Tripathi and Mishra, 2017). With India's population of 1.4 billion growing at a rate of 2.23% per year (UNDESA, 2022), wheat will play a major role in ensuring food supply meets the growing demand (Tripathi and Mishra, 2017). However, India's croplands are exposed to particularly high $O_3$ concentrations ($[O_3]$) with hotspots occurring across the Indo-Gangetic Plains (IGP) (Roy et al., 2008). The 8-hour daily mean $O_3$ concentrations often reach up to 100 ppb in hotspots during the Rabi crop growing season (October to April) and are therefore a significant threat to India's wheat productivity (Roy et al., 2009). Currently, there are no air quality standards in India to protect crops from surface $O_3$, and emissions of $O_3$ precursors are forecast to continue to rise well into the 21$^{st}$ century, driven by persistent growth in industries, including mining and petroleum industries, vehicular traffic and agricultural activities (Ghude et al., 2014; Sharma et al., 2019; Yadav et al., 2019). Ozone distribution varies spatially and temporally but the IGP often experiences high levels due to the long-distance transport of $O_3$ and its precursors from urban, industrial or power generation centres located across northern India (Singh and Agrawal, 2017).

Ozone damages crops when it diffuses into the intracellular airspace of the leaf via the stomata which triggers a cascade of metabolic and physiological responses resulting in reduced carbon assimilation, premature leaf senescence and visible injury. Together, these effects can lead to reductions in overall yield and quality (Emberson et al., 2018). Since $O_3$ damage relies on stomatal $O_3$ flux (i.e., $O_3$ dose), the scale of damage caused by ambient $[O_3]$ varies with stomatal conductance. Stomatal conductance is determined in the short-term, by environmental factors that trigger the closure of the stomata, and in the long term, by adaptations to climate change i.e., reduced stomatal density (Emberson et al., 2018). Two other factors also influence a crop's vulnerability to $O_3$ dose; its detoxification ability and the signal transduction pathway, which regulates the response of cells to the increased oxidative load caused by $O_3$ (Ainsworth et al., 2008; Kangasjärvi et al., 2005).

It is widely acknowledged that stress conditions including elevated levels of carbon dioxide ($CO_2$), heat and water vapour pressure deficit (VPD) and soil water deficit (all of which may be associated with climate change) decrease stomatal conductance, thus reducing $O_3$ flux in wheat and potentially ameliorating $O_3$ damage to the photosynthetic apparatus (Feng et al., 2008). In addition, several studies have found modern cultivars are more $O_3$-sensitive due to selection for enhanced gas exchange, which could counteract their natural adaptation of lower stomatal conductance resulting from the changing climate. Pleijel et al. (2006) and Yadav et al. (2020) observed greater $O_3$-related yield loss in a modern wheat cultivar, HD-3118, bred for a higher yield than HUW-234, which was attributed to the cultivar's higher stomatal density and conductance. Climate change is expected to increase the use of drought-resistant cultivars which can maintain higher stomatal conductance under drought conditions, this will likely increase crop sensitivity to $O_3$ (Emberson et al., 2018). Eliminating the use of cultivars with higher stomatal conductance is unlikely to improve productivity because, on a broader scale, yield losses due to water stress outweigh those from $O_3$ (Emberson et al., 2018). However, in major wheat-producing states like Uttar Pradesh, where irrigation

is widespread (Zaveri and Lobell, 2019) yield losses due to O3 may outweigh those due to water stress, hence the use of cultivars with a lower stomatal conductance may be beneficial.

Khan and Soja (2003) found that well-irrigated wheat plants (i.e. with a 75% soil water capacity (SWC)) suffered grain yield losses of up to 39% when exposed to accumulated $O_3$ concentrations over a threshold of 40 ppb (AOT40) of ~25ppm/h. Under severe moisture deficit (35% SWC), no $O_3$-related yield loss was observed as $O_3$ uptake was reduced by up to 90%. However, the grain yield of water-stressed wheat was significantly less than well-watered wheat. In a similar study by Harmens et al. (2019) on wheat in Africa, grain yield loss due to $O_3$ exposure was greater in well-watered plants than in crops that received reduced irrigation suggesting controlled irrigation as a management tool to reduce $O_3$ impact. Whilst drought reduces yields at all stages of development, drought stress during anthesis and grain-filling cause the greatest yield reductions (Farooq et al., 2014). Additionally, anthesis and grain filling is when wheat is most sensitive to $[O_3]$ and is the period of time when the $[O_3]$ are highest during the Indian growing season (Gelang et al., 2000; Pleijel et al., 1998; Rathore et al., 2023). Several experimental studies have investigated the interaction between $O_3$ and drought stress in wheat. While some studies have observed no significant interactions between increased $[O_3]$ and water stress (Broberg et al., 2023; Fangmeier et al., 1994), others have observed an interaction. Ghosh et al., (2020) observed an additive effect of $O_3$ and drought stress, with a greater reduction in grain yield when both stressors occurred simultaneously due to the reduction in nutrient uptake and assimilation. As a result of these contrasting findings, it is evident the trade-offs between $O_3$ exposure and water stress require further study. Irrigation has the potential to maximise $O_3$-stress by providing conditions likely to enhance stomatal conductance such as plentiful soil and leaf water, transpirational cooling and low leaf-to-air VPD. Irrigation is widespread across the IGP, particularly in the states of Punjab, Haryana and Uttar Pradesh; the area irrigated as a percentage of the total area of wheat was 99.1%, 99.9% and 99% respectively in 2018-19 (Ministry of Agriculture & Farmers Welfare, 2022), which means that current wheat crop management practices are likely to enhance sensitivity to $O_3$. Modifying irrigation practices has been suggested as a strategy to reduce $O_3$ damage, but caution is needed to avoid introducing water stress, which could also negatively affect yield (Harmens et al., 2019; Teixeira et al., 2011). Irrigation has additional benefits and has often been implemented to offset heat-related yield losses which occur when temperatures exceed 35°C (Zaveri and Lobell, 2019). However, studies suggest that sustainable use of India's future groundwater availability with current irrigation practices would mitigate less than 10% of the climate change impact on crop yield (Fishman, 2018). Additionally, water for irrigation purposes is limited; for example, Zaveri et al. (2016) found Uttar Pradesh will lack scope for further increasing irrigation as groundwater depletion escalates due to climate change and increased unsustainable water demand. With irrigation accounting for up to 90% of India's total water demand, water efficiency in agriculture is a priority in the IGP to achieve better environmental and economic performance (Fischer et al., 2007; Wada et al., 2013). Here, we explore the interplay between $O_3$- and water-stressed induced yield losses which will help inform whether water efficiencies could also provide some benefits in terms of the decreased sensitivity of staple crops to $O_3$.

There have been an increasing number of studies exploring the effect of $O_3$ on wheat yields using a cumulative stomatal $O_3$ flux metric ($POD_Y$; phytotoxic $O_3$ dose over a flux threshold Y) which accounts for the stomatal response to environmental

conditions and plant growth stages that alter $O_3$ uptake. By comparison, concentration-based exposure metrics such as AOT40
only account for atmospheric $[O_3]$ which may be decoupled from $O_3$ uptake under environmental conditions that limit stomatal
conductance; they also omit $O_3$ below 40 ppb which are known to be capable of causing damage (CLRTAP, 2017; Emberson
et al., 2018). Mills et al. (2018a) estimated an $O_3$-induced yield loss that incorporated the effects of irrigation to be in the range
of 15-20% for wheat growing in Uttar Pradesh between 2010-2012 using $POD_3IAM$ (POD above 3nmol m$^{-2}$s$^{-1}$, parameterised
for integrated assessment modelling) (CLRTAP, 2017). This parameterisation was based on European wheat cultivars and a
broad-scale assessment of India's wheat growing season with the $POD_3IAM$ metric being applied according to the formulations
of the Deposition of Ozone for Stomatal Exchange (DO$_3$SE) model (Büker et al., 2012; CLRTAP, 2017; Emberson et al.,
2000a, 2018). An alternative metric used in estimations of stomatal $O_3$ flux effects on crops is POD6SPEC, which is the
species-specific phytotoxic $O_3$ dose above 6 nmol m$^{-2}$ s$^{-1}$. This metric is better suited for local-regional risk assessments
(CLRTAP, 2017).
Application of the stomatal $O_3$ flux method allows exploration of the relative effects of both $O_3$ and water stress on yield since
the DO$_3$SE model also estimates water vapour fluxes from which potential and actual evapotranspiration can be calculated
(Büker et al., 2012). In this study, we parameterise the DO$_3$SE 3.1.0 version model for two late-sown Indian wheat cultivars
for the estimation of POD$_6$SPEC metric. We apply the Weather Research and Forecasting model with Chemistry (WRF-Chem)
(Grell et al., 2005) to obtain $[O_3]$ and climate variable data for Varanasi, Uttar Pradesh. We compare $O_3$ effect yield losses
(using the wheat grain yield flux-effect relationship (CLRTAP, 2017) with yield losses due to water stress based on yield
responses to the ratio of actual vs potential evapotranspiration (cf. FAO (2012)). This modelling set-up allows us to explore
the relative magnitude of yield losses from $O_3$ and water stress; how these are likely to change in the future and the relative
sensitivities of older vs more recently released Indian cultivars to damage from $O_3$ pollution.
**2 Methods**
For this study, Varanasi was selected as the study area due to (i) its location in the important wheat-growing IGP and, (ii) the
availability of observed crop and $O_3$ data.
**2.1 Experimental data**
Experimental data from 2016-2018 were obtained for two late sown Indian spring wheat (*Triticum aestivum* L.) cultivars
grown at the Botanical Garden, Banaras Hindu University (BHU), Varanasi (25°16′ N, 82°59′ E; 81.0m above sea level).
HUW-234 (released in 1986 by BHU, Varanasi) and HD-3118 (released in 2014 by IARI, New Delhi) were selected based on
their heat tolerance and extensive cultivation in the North East Plain Zone of India (Joshi et al., 2007; Yadav et al., 2019). The
recently released cultivar, HD-3118 is more high-yielding (6.64 tons ha$^{-1}$) compared to HUW-234 (4.5–5 tons ha$^{-1}$), most
likely due to its enhanced capacity for gas exchange. This enhanced capacity for gas exchange is the likely reason for the HD-
3118 cultivar having a greater sensitivity to $O_3$ than the HUW-234 cultivar (Yadav et al., 2020).

## 2.2 Modelled data

Hourly meteorological and O3 data for the Varanasi grid box (45 km x 45 km horizontal resolution) were obtained by running WRF-Chem v.3.8.1 for years 1996-2005 (considered the 'recent past' climate) and 2046-2055 using both RCP4.5 and RCP8.5 climate scenarios. The 45 km resolution model domain is the same as in Daloz et al. (2021), and the meteorological initial and boundary conditions come from global climate model simulations with the Community Earth System Model (CESM) v.1.0.4 (Gent et al., 2011), documented in Hodnebrog et al. (2019). The WRF-Chem simulations are set up with the RADM2 gas-phase chemistry scheme (Stockwell et al., 1990) and $O_3$ precursor emissions are from Lamarque et al. (2010) for the historical period and from Lamarque et al. (2011) for the future RCPs. Global mean $CO_2$ mixing ratios (ppm) for 1996-2005 were obtained from NASA (using Tans and Conway (no date) for 1983-2003 and Conway (no date) for 2004-2007). For future scenarios, RCP4.5 and RCP8.5 [$CO_2$] for 2046-2055 were acquired (Meinshausen et al., 2011). WRF-Chem with the RADM2 chemical mechanism has been found to reproduce diurnal average $O_3$ over India in February-May relatively well, while noontime $O_3$ concentrations show considerable differences between simulations with various emission inventories (Sharma et al., 2017).

The RCP climate scenarios are selected to provide a range of climate and pollution futures for India from which a consequent range in yield responses can be estimated. RCPs are possible greenhouse gas (GHG) emission pathways designed to aid research into climate change impacts (Riahi et al., 2011). RCP8.5 is a very high baseline, representing the highest GHG emission pathway in a 'business as usual' scenario resulting in a radiative forcing of 8.5 $Wm^{-2}$ at the close of the 21st century; equivalent to 1370 ppm [$CO_2$] (He and Zhou, 2015; Riahi et al., 2011). RCP4.5 is a medium stabilisation scenario where global climate policy values the role of natural carbon sequestration and land use, resulting in a radiative forcing target of 4.5 $Wm^{-2}$ (650ppm [$CO_2$] equivalent) for 2100 (Riahi et al., 2011; van Vuuren et al., 2011).

These data provided input to the $DO_3SE$ model which was used to simulate stomatal $O_3$ flux values and water stress characteristics for the two cultivars for each year. The modelled climate, $O_3$ and $CO_2$ data for the recent past climate and both RCP scenarios are summarised in Table 1. The modelled temperature data are on average 1.3°C warmer in the RCP4.5 scenario and 1.9°C warmer in the RCP8.5 scenario than the recent past modelled climate.

**Table 1: Modelled climate, [$O_3$] and [$CO_2$] data for the Varanasi grid box, used for the recent past climate and both future RCP scenarios (expressed as the range of 24-hour mean values, value in brackets indicates mean). The length of growing season is shown in days over two years and can be visualised in Fig. 2.**

| Parameter | Recent past **climate** (1996-2005) | RCP4.5 (2046-2055) | RCP8.5 (2046-2055) |
|---|---|---|---|
| Temperature (°C   ) | 17.4-20.6 (18.9) | 19.1-21.3 (20.2) | 19.6-21.8 (20.8) |
| VPD (hPa) | 8.3-14.4 (11.0) | 12.5-17.5 (14.5) | 11.1-18.1 (15.1) |

| | | | |
|---|---|---|---|
| Precipitation (total over growing season; mm) | 72.4-393.4 (235.5) | 35.1-184.4 (101.7) | 0.93-234.0 (92.5) |
| [$O_3$] (24 hour mean; ppb) | 47.1-50.6 (48.6) | 57.9-62.2 (60.5) | 54.6-63.0 (59.7) |
| [$CO_2$] (ppm) | 362.6-379.5 (370.7) | 476.3-498.5 (487.6) | 518.6-570.5 (543.9) |
| Growing season (Days over 2 years) | 339-468 | 339-466 | 339-473 |

**2.3 Model formulation - O3 induced yield loss estimates**

The DO$_3$SE 3.1.0 version model (https://www.sei.org/projects-and-tools/tools/do3se-deposition-ozone-stomatal-exchange/) was used to estimate stomatal $O_3$ flux and subsequent $O_3$-induced yield loss for wheat. DO$_3$SE is a dry deposition model which takes into account the influence of climatic, soil and plant factors on stomatal conductance to estimate stomatal $O_3$ flux and determine the accumulated stomatal $O_3$ uptake during a specified growth period; POD$_Y$ (CLRTAP, 2017). The stomatal conductance ($g_{sto}$) multiplicative algorithm Eq. (1) used in DO$_3$SE estimates hourly $g_{sto}$ to $O_3$ by modifying a species-specific maximum $g_{sto}$ ($g_{max}$) according to environmental variables and is described in Emberson et al. (2000a, b).

$$g_{sto} = g_{max} \times [min(f_{phen}, f_{O3})] \times f_{light} \times max\{f_{min}, (f_{temp} \times f_{VPD} \times f_{sw})\} \quad [1]$$

where $g_{sto}$ and $g_{max}$ are expressed as mmol $O_3$ m$^{-2}$ PLA s$^{-1}$. The factors $f_{phen}$, $f_{O3}$, $f_{light}$, $f_{temp}$, $f_{VPD}$, $f_{sw}$ and $f_{min}$ represent the influence of phenology, [$O_3$], light, air temperature, VPD, soil water potential and minimum $g_{sto}$ and are expressed in relative terms as a proportion of $g_{max}$ (so have a value between 0-1). Functions describing these factors for environmental conditions are described in CLRTAP (2017) based on European wheat varieties; for $f_{sw}$ (and to simulate $g_{sto}$ for rainfed wheat) we assume a linear relationship between a relative $g_{sto}$ of 1 and $f_{min}$ at soil water potentials (SW) of -0.3 and -1.1 MPa (Ali et al., 1999; Morgan, 1984). To simulate the $g_{sto}$ of irrigated wheat we simply assume that $f_{sw}$ is always equal to 1.

Stomatal $O_3$ flux ($F_{st}$; nmol m$^{-2}$ PLA s$^{-1}$) was calculated using Eq. (2).

$$F_{st} = c(zi) \times g_{sto} \times \frac{rc}{(rb+rc)} \quad [2]$$

where $c(zi)$ is [$O_3$] at the top of the canopy height $i$ (m), $rc$ and $rb$ represent leaf surface and quasi-laminar leaf boundary layer resistances respectively, based on leaf dimension and wind speed (CLRTAP, 2017).

The species-specific POD$_Y$ (POD$_Y$SPEC) is estimated for the wheat accumulation period according to Eq. (3).

$$POD_Y SPEC = \sum \left[ (F_{st} - Y) \times \left( \frac{3600}{10^6} \right) \right] \quad [3]$$

where $Y$ (nmol $O_3$ m$^{-2}$ PLA s$^{-1}$) is subtracted from $F_{st}$ (in nmol m$^{-2}$ PLA s$^{-1}$) when $F_{st}>Y$, during daylight hours; this $Y$ value represents the assumed detoxification capacity of wheat to $O_3$ flux. The value is then converted to hourly fluxes by multiplying by 3600 and to mmol by dividing by $10^6$ to give $POD_Y SPEC$ in mmol $O_3$ m$^{-2}$ PLA (CLRTAP, 2017). A $Y$ value of 6 nmol m$^{-}$

$^2$ PLA s$^{-1}$ was used based on values for European wheat (CLRTAP, 2017). The resulting $POD_6SPEC$ values were used to
estimate the percentage grain yield (relative to 100% grain yield under pre-industrial $O_3$ conditions) based on the dose-response
relationship in Eq. (4).
$\%Grain\ Yield = 100.3 - (3.85 \times POD_6SPEC)$         [4]
The relationship in Eq. (4) is taken from CLRTAP (2017)     where relative grain yields from 5 wheat cultivars in 4 European
countries were regressed against the POD$_6$SPEC value.
**2.4 Model formulation - water stress induced yield losses**
The DO$_3$SE 3.1.0 model was also used to model the effect of water stress on yield through the provision of estimates of
potential ($ET_m$) and actual ($ET_a$) evapotranspiration following the DO$_3$SE model algorithms used to estimate soil-plant-
atmosphere cycling of water described in Büker et al. (2012). These DO$_3$SE algorithms essentially estimate the total loss of
soil water through $ET_a$ (and the equivalent $ET_m$) using the method of Shuttleworth and Wallace (1985) modified to incorporate
the atmospheric, boundary layer and stomatal resistances to water vapour flux as calculated within DO$_3$SE. Resistances are
scaled from leaf to canopy using LAI and upscaling methods described in Büker et al. (2012). LAI is modelled to vary over
the course of the wheat growing season between a value of 0 and 3.5 m$^2$/m$^2$ (consistent with average maximum LAI values
frequently found across the IGP region as observed from satellite data (Nigam et al., 2017). The DO$_3$SE soil moisture module
was developed based on the Penman-Monteith model of actual evapotranspiration ($Et$a), which is described in Eq. (5) (Büker
et al., 2012; Montieth, 1965; Shuttleworth and Wallace, 1985):
$$Eta = \frac{\Delta\,(\Phi n - G) + \rho_a c_p \left(\frac{D}{R_{bH_2O}}\right)}{\lambda \left\{ \Delta + \gamma \left(1 + \frac{R_{stoH_2O}}{R_{bH_2O}}\right) \right\}}$$         [5]
where $\Delta$ is the slope of the relationship between the saturation vapour pressure and temperature, $\Phi n$ is the net radiation at the
top of the canopy, $G$ is the soil surface heat flux, $\rho_a$ is the air density, $c_p$ is the specific heat of air, $D$ is the vapour pressure
deficit of air, $R_{bH_2O}$ is the canopy boundary layer resistance to water vapour exchange, $R_{stoH_2O}$ is the stomatal canopy
resistance to the transfer of water vapour (the inverse of stomatal conductance to water vapour), $\gamma$ is the psychrometric constant,
and $\lambda$ is the latent heat of vaporisation.
The effect of $ET_a$ (and hence water-stress) on wheat yield was estimated according to the relationship between relative yield
and the corresponding relative evapotranspiration ($Et$) described in Doorenbos and Kassam (1979) for spring wheat. When a
crop is not water-stressed, $ET_a$ is equal to $ET_m$ however in drought conditions, $ET_a < ET_m$ (Yao, 1974). The $ET_a$ and $ET_m$ values
produced by DO$_3$SE were used in Eq. (6).
$1 - \frac{Y_a}{Y_m} = K_y \left(1 - \frac{ET_a}{ET_m}\right)$         [6]

where $Y_a$ is the actual relative grain yield and $Y_m$ is the potential relative grain yield. $K_y$ is the crop-specific yield response factor assumed to be 1.15 for the whole growing season, in accordance with the value for spring wheat from the FAO (Steduto et al., 2012).

**2.5 Model parameterisation**

The DO₃SE model was parameterised for the HD-3118 and HUW-234 cultivars by (Yadav et al. (2021) using data from a series of O₃ exposure experiments at the Banaras Hindu University, Varanasi, Uttar Pradesh. For this study, we use the same parameterisation except for the $f_{phen}$ term (which we allow to vary as a function of effective temperature sum (ETS) during the growing season) and the inclusion of the $f_{O3}$ term (which accounts for O₃ inducing early onset senescence). The parameterisations used for both cultivars are given in Table 2.

**Table 2: Parameterisation of the DO₃SE model for POD₆SPEC for wheat flag leaves for Indian bread wheat (Triticum aestivum L.) cultivars. European bread wheat parameters reported by CLRTAP (2017) have been included for comparative purposes. Parameters are highlighted where there are differences between Indian and European cultivars.**

| | | Bread wheat cultivar parameterisation - POD₆SPEC | | |
|---|---|---|---|---|
| **Parameter** | **Units** | Indo-Gangetic Plains | | Atlantic, Boreal, Continental bread wheat (CLRTAP, 2017) |
| | | HUW-234 | HD-3118 | |
| $g_{max}$ | mmol O₃ m⁻² PLA s⁻¹ | 500 | 521 | 500 |
| $f_{min}$ | fraction | 0.13 | 0.13 | 0.01 |
| light_a | - | 0.011 | 0.011 | 0.011 |
| $T_{min}$ | °C | 12 | 12 | 12 |
| $T_{opt}$ | °C | 26 | 26 | 26 |
| $T_{max}$ | °C | 40 | 40 | 40 |
| $VPD_{max}$ | kPa | 3.2 | 3.2 | 1.2 |
| $VDP_{min}$ | kPa | 4.6 | 4.6 | 3.2 |

| | | | | |
|---|---|---|---|---|
| $\sum VPD_{crit}$ | kPa | 16 | 16 | 8 |
| $PAW_t$* | % | 50 | 50 | 50 |
| $f_{O3}$ | $POD_0$ mmol $O_3$ m$^{-2}$ PLA s$^{-1}$ | 14 | 14 | 14 |
| Leaf dimension | cm | 2 | 2 | 2 |
| Canopy height | m | 1 | 1 | 1 |
| $f_{phen\_a}$ | fraction | 0.3 | 0.3 | 0.3 |
| $f_{phen\_e}$ | fraction | 0.7 | 0.7 | 0.7 |
| $f_{phen\_1\_ETS}$ | °C day | -616.6 | -553 | -200 |
| $f_{phen\_2\_ETS}$ | °C day | 0 | 0 | 0 |
| $f_{phen\_3\_ETS}$ | °C day | 621.5 | 553 | 100 |
| $f_{phen\_4\_ETS}$ | °C day | 182.75 | 238 | 525 |
| $f_{phen\_5\_ETS}$ | °C day | 959 | 1000 | 700 |

*$PAW_t$ is the threshold for plant available water (PAW) in mm above which stomatal conductance is at a maximum

The ETS model (see Eq. 7) was calibrated using experimental data for HUW-234 and HD-3118 that provided the timing (as
day of year) of key crop development stages (sowing, emergence, flag leaf emergence, fully expanded flag leaf, start of seed
setting, start of senescence and harvest) for both cultivars for 3 years (2016 to 2018 inclusive). Corresponding 3-hourly
temperature data were used to estimate daily mean temperature from which ETS values could be determined according to
Eq. (7).
$$ETS = \sum (T_i - T_b) \qquad [7]$$
Where $T_i$ is the mean daily temperature and $T_b$ is the base temperature assumed 0°C for wheat. This is equivalent to the
method of thermal time accumulation recommended by CLRTAP (2017) and assumes that there is no upper threshold
temperature for phenology and that thermal time increases linearly across the entire temperature range.
The ETS components of the $f_{phen}$ function from flag leaf emergence were estimated by assuming that $f_{phen}1\_ETS$ and
$f_{phen}3\_ETS$ together were equivalent to thermal time equally divided between the emerging flag leaf and seed setting. This
precaution ensured $f_{phen}$ was not allowed to decrease too early; that $f_{phen}5\_ETS$ less $f_{phen}4\_ETS$ was equivalent to the
thermal time at seed setting less the thermal time at flag leaf emergence and that $f_{phen}1\_ETS$ and $f_{phen}5\_ETS$ together were
equivalent to the thermal time at harvest less the thermal time at flag leaf emergence; these basic assumptions allowed the
derivation of $f_{phen}\_ETS$ parameters 1-5 given in Table 2. Supplement Figure S1 gives an indication of the year-to-year
variability in the timing of these key growth stages used to parameterise the $f_{phen}$ function. This $f_{phen}$ function is
subsequently used to represent the phenological influence on $g_{max}$ and to define the seasonal accumulation period for
POD$_Y$SPEC (see also CLRTAP (2017)). Parameterisation of the $f_{phen}\_ETS$ model shows little difference in phenology
between these cultivars, although HUW-234 had a greater range in dates of flag leaf emergence and fully expanded flag leaf.
Seed setting and the start of senescence occurred ~3 days earlier in HD-3118 than HUW-234. The Indian cultivar $f_{phen}\_ETS$
values differ from the European Continental bread wheat values (also shown in Table 2). In part, this is related to the
precautionary approach taken in defining the length of the period during which $f_{phen}$ will equal 1 to ensure we capture the
period when O$_3$ may be taken up by the stomata in the absence of growth stage data more specific to the $f_{phen}\_ETS$ stages.
The resulting $f_{phen}\_ETS$ parameterization suggests that Indian cultivars take more thermal time to reach mid-anthesis and
less thermal time between the start of senescence and harvest than would bread wheat from the European region.
**2.6 Model runs**
DO$_3$SE 3.1.0 model runs were made for each cultivar described in Table 2 using the WRF-Chem modelled O$_3$ and met data
for 1996-2005 which was assumed to represent the recent past climate. The DO$_3$SE model runs were repeated for future
scenarios using the WRF-Chem modelled O$_3$ and meteorological data for 2046-2055 based on the two scenarios; RCP4.5
and RCP8.5 to explore the influence of changes in climate and O$_3$ precursor emissions on O$_3$ uptake. We assume a sowing
date     of early November since October to December represent the main sowing months of wheat across the most
productive wheat growing states in the IGP (Lobell et al., 2013).
**3 Results and discussion**
**3.1 Phenology and stomatal O3 uptake**
Accurate modelling of the growing season and the $f_{phen}$ period in relation to the prevailing O$_3$ climate is crucial for realistic
estimates of O$_3$ damage to wheat. The ETS model for late-sown cultivars is variable in its ability to simulate key growth
stages between years and cultivars. For each growth stage, the minimum and maximum ºCday values between years are 63 to
426ºCday for HUW-234 and 63 to 317ºCday for HD-3118 respectively. Given that the mean daily temperature during the
Indian wheat growing season is ~25ºC this would suggest the ETS model may have a maximum error of 17 and 13 days for
HUW-234 and HD-3118 respectively. These values are likely at the high end of the uncertainty range as temperatures
increase during the growing season and the greatest uncertainty was found for the flag leaf emergence and fully expanded
flag leaf growth stages. The inclusion of the 'emerging flag leaf' in the $f_{phen}$ period helps to capture the full period when the
flag leaf may be vulnerable to $O_3$ as a precaution given the uncertainty in the ETS model defining the timing of the period
from full flag leaf expansion and senescence.

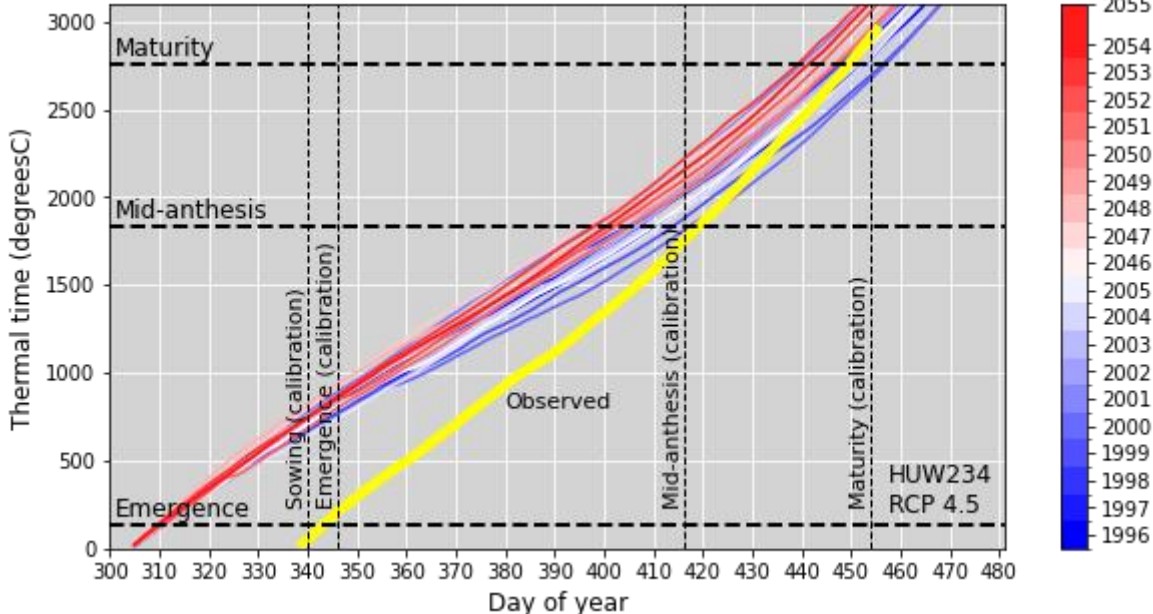


**Fig. 1: The evolution of ETS and associated growth stages for the observed (2016-18) climate (to which the ETS model is calibrated**
**with a sowing date of 5th Dec) and the WFR-Chem modelled recent past   (1986-2005) and future (2046-2055; RCP4.5 and 8.0)**
**climates (for which the model is applied with a sowing date of 5th Nov) for the HUW-234 cultivar.**
When the parameterisation was applied to the WRF-Chem modelled 1996-2005 climate temperature data, with a sowing date
of 5th November, maturity is simulated to occur around the end of March (consistent with our observed maturity date of the
late sown variety under the relatively high temperatures for years 2016-2018 used for parameterisation) – see Fig. 1. Thus
our $f_{phen}$ parameterisation, when using standard, early November, sowing dates gives realistic maturity dates for IGP grown
wheat when used with recent past    WRF-Chem modelled data (Fig. 2). Since the WRF-Chem data are consistent between
climate periods (i.e., 1996-2005 and 2046-2055) they can be deemed to provide a means of comparing the relative effect of
changes in temperature on the growing season, $O_3$ uptake and the evolution of soil moisture deficit.
The empirical data used for model parameterisation collected in years 2016-18, consistently produced higher temperatures
than the WRF-Chem model-based meteorological data, which was collected from 1996-2005 (Fig. 3a). This could be in part
due to climate change; average air temperatures in India for 2016, 2017 and 2018 were in the top ten on record since 1901
(ESSO, 2019). However, on average, 2016-18 was only +0.72°C, +0.55°C and +0.41°C warmer than the 1981-2010 annual
air temperature average respectively (Earth System Science Organisation et al., 2019), therefore it is likely that uncertainties
in the modelled values caused the greater part of these discrepancies in temperatures. Since the WRF-Chem model at this
resolution may not consider some urban heat island effects, a finer model resolution may have led to better agreement with
observations for this urban site. Despite this, the nature of the ETS model is that it can provide comparative estimates of the
influence of temperature profiles on the timing and length of the growing season. Fig 3b shows a similar comparison
between WRFChem modelled $O_3$ concentrations (provided as 5 year mean hourly values with absolute minimum and
maximum bounds also shown) and the ambient air (AA) $O_3$ concentration data for the 2016-2017 wheat growing season of
the $O_3$ experiment. This shows that the WRFChem modelled past climate (1996-2005) data is within range, but at the lower
end, of the 2016-17 $O_3$ experimental data as would be expected given the increase in $O_3$ precursor emissions over the past
few decades.

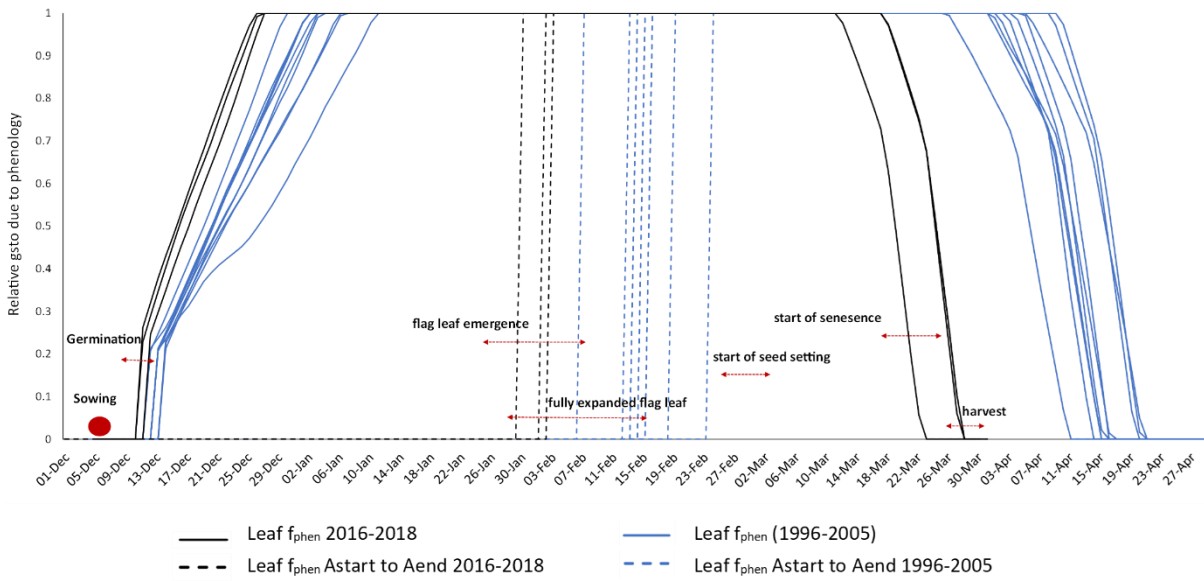


Leaf $f_{phen}$ 2016-2018          Leaf $f_{phen}$ (1996-2005)

Leaf $f_{phen}$ Astart to Aend 2016-2018     Leaf $f_{phen}$ Astart to Aend 1996-2005

**Fig. 2: The ETS model parameterised for HUW-234 based on observed recent past     temperatures (2016-2018; black). The ETS**
**model for the modelled recent past     temperatures (years 1996-2005) are in blue. The range of observed dates of sowing,**
**germination, flag leaf emergence, fully expanded flag leaf, start of seed setting, start of senescence and harvest from the**
**experimental data (Agrawal, pers. comm.) are marked with arrows. Astart to Aend represent the start of anthesis to the end of**
**anthesis.**











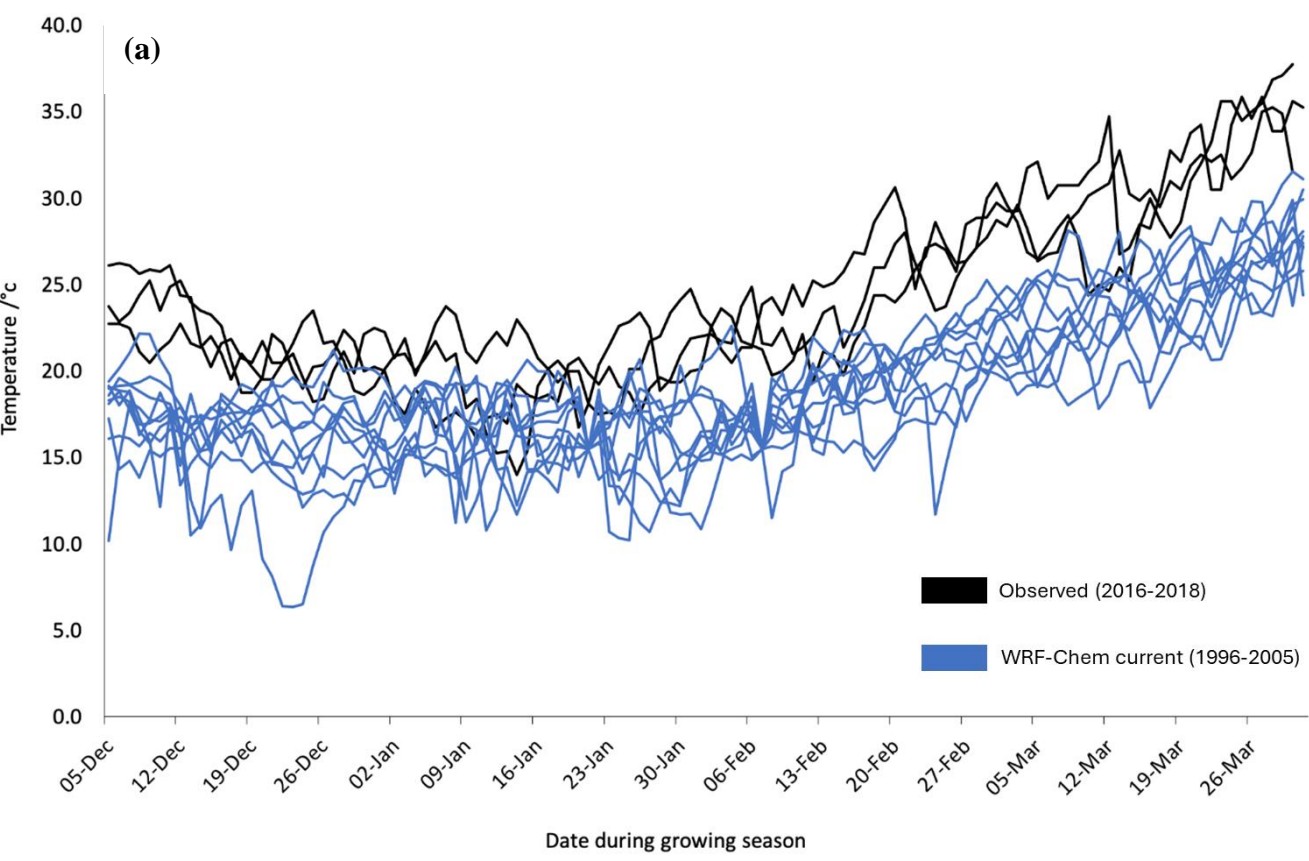


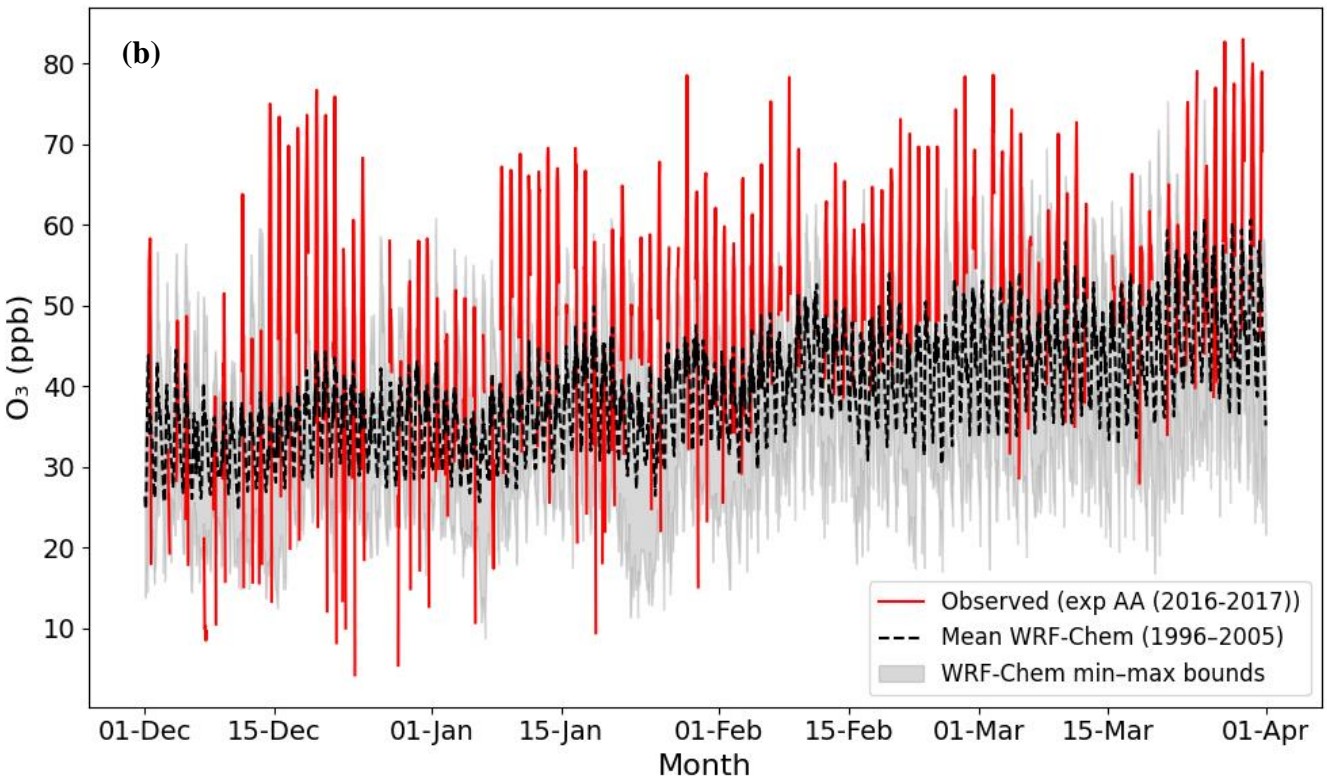


**Fig. 3: Seasonal profiles of (a)    surface temperatures    over the growing season for observed    data (2016-18) and WRF-Chem modelled data for the recent past    climate (1996-2005)    and (b) O3 concentration over the growing season for observed data (exp AA (experiment ambient air) for Dec 2016 to Mar 2017 and 10 year (1996 - 2005) annual mean  WRF Chem modelled O3 concentration data with associated absolute minimum and maximum O3 concentrations bounds also shown.**

**The DO₃SE wheat stomatal O3 flux model has been evaluated against wheat $g_{sto}$  data (the primary determinant of stomatal ozone flux) collected under experimental conditions in Ostad, Sweden (Pleijel et al. 2007) and found to perform well (with an R2 value of 0.83 for a regression of observed against modelled $g_{sto}$). The DO₃SE model has also been extensively evaluated for a number of crops at locations around the world (as reported in Tuovinen et al. (2004) for wheat growing near Cumono novo in Italy and Emmerichs et al. (2025) for wheat growing near Grignon in France). These evaluations rely on total O₃ flux and deposition measurements (since they use O₃ flux tower data) or water vapour flux measurements and thereby test whole canopy fluxes rather than the representative upper leaf stomatal flux required for PODy calculations. However, the combination of these evaluation methods focussing on both leaf level $g_{sto}$ and canopy level O₃ flux together provide confidence in the predictive abilities of the DO₃SE model.**

### 3.2 Effect of O₃ stress on the yield benefits of irrigation

Water-stress induced yield loss under rainfed conditions modelled under the climate scenario for 1996-2005 was found to exceed O₃-relattive    yield loss (O₃.RYL) under irrigated conditions for the majority of the 10 years investigated. Under this climate, rainfed conditions produced a mean water-stress relative    yield loss (WS-RYL) of 13.3% for HUW-234, with a

range of 2.8-31.3% (Fig. 4a). Under rainfed conditions, mean $O_3$-RYL was projected to be negligible (0.6%), significantly
lower than the mean $O_3$-RYL when no water-stress is assumed under irrigation (10.7% with a range of 4.8-15.4%). This
demonstrates the importance of irrigation for wheat production in India and highlights the substantial influence on the yield of
$O_3$ for irrigated wheat.
$O_3$-RYL under irrigated conditions exceeded WS-RYL in 80% of the 10 years investigated in the RCP4.5 scenario (Fig. 4b).
This highlights how $O_3$ stress negates some of the increased productivity that arises from reducing water stress through
irrigation. In the RCP8.5 scenario, WS-RYL under rainfed conditions exceeded $O_3$-RYL under irrigated conditions in all but
one year (2051), when precipitation totaled 234.0mm for the growing season (Fig. 4c). In this scenario, precipitation during
the growing season ranges from 0.9-234.0mm, with a mean of 94±82.97mm, and as a result the WS-RYL fluctuates within the
10 years.
Whilst irrigation has played an important role in increasing yields for India's wheat, these results show that $O_3$ is likely to
negate some of the yield benefits of irrigation. Based on the results of simulations of future climates, irrigation will have less
of an effect on yield increases as [$O_3$] levels rise.

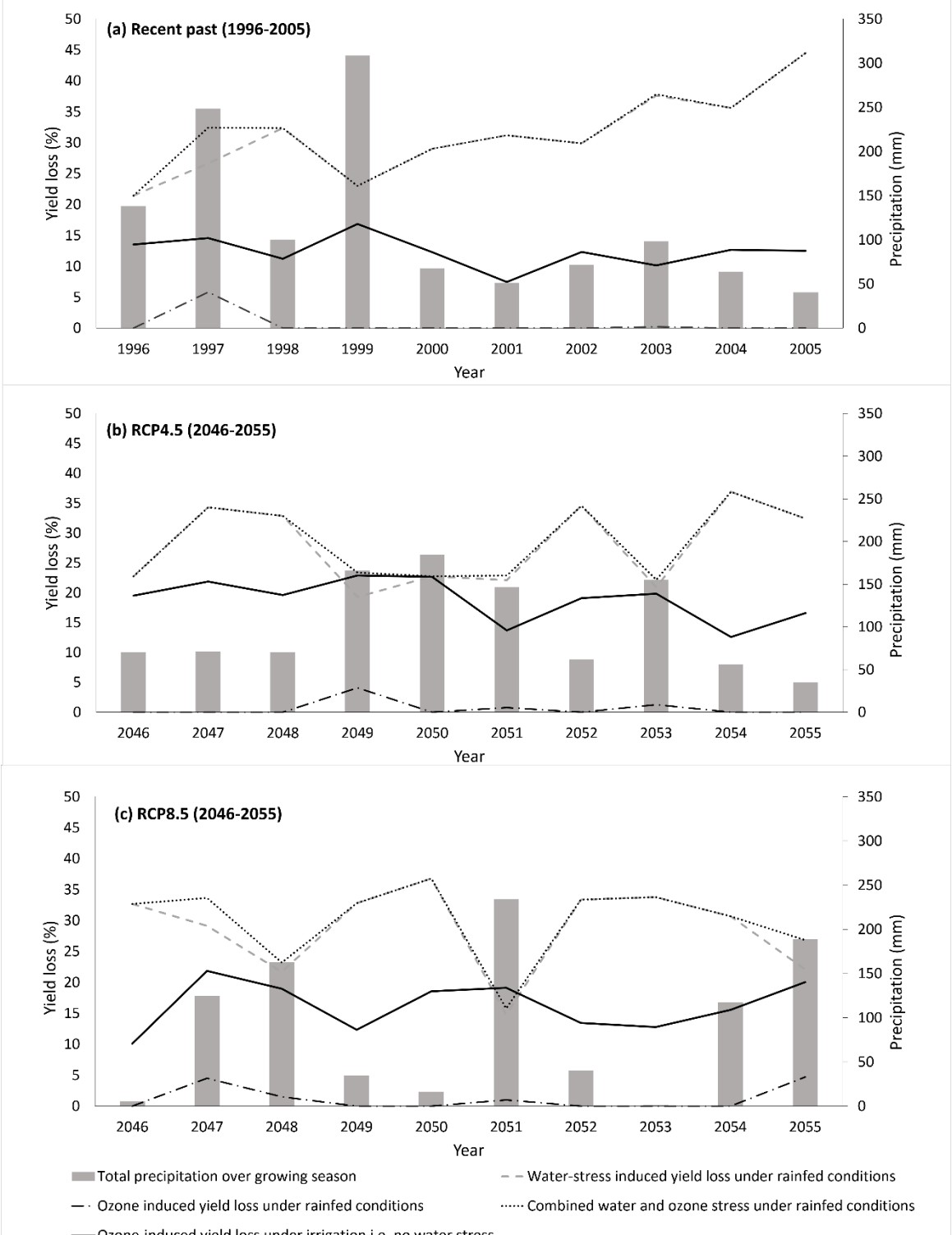


**Fig. 4: The water-stress and ozone-related relative yield loss modelled for HUW-234 under rainfed conditions (water stress) and with no water stress ($f_{sw}$ set to 1 in DO$_3$SE) for (a) the recent past      climate 1996-2005; (b) RCP4.5 scenario 2046-2055; (c) RCP8.5 scenario 2046-2055.**

The mean total precipitation for the growing season under the 1996-2005 climate scenario was higher than the median values in the RCP4.5 and RCP8.5 scenarios for 2046-2055, meaning Uttar Pradesh's crops will receive less rainfall in the future. In addition, the RCP8.5 scenario had a larger interquartile range (IQR) of 131.0mm than the 1996-2005 climate (54.8mm) and the smallest lower quartile (22.3mm), demonstrating less and more irregular precipitation in the future climate. Whilst RCP4.5 was less extreme; it had a larger IQR of 90.1mm. This irregularity and increased risk of low precipitation over the growing season demonstrates the continuing importance of irrigation for wheat productivity. In all modelled climate scenarios, water stress tends to be a much greater threat to crop yields than O$_3$ and therefore, some level of irrigation is crucial for sustained wheat productivity in India. However, these findings clearly show that O$_3$ is a limiting factor to yield under irrigated conditions meaning that the full potential benefit of irrigation is not being realised and hence will lead to inefficiencies in the use of irrigation water. Further research should be carried out to find the 'sweet spot' for irrigation, that will minimise O$_3$ stress without inducing water stress, to practice more responsible water management.

Future studies should investigate how short, sharp high O$_3$ periods could be mitigated with temporary reductions in irrigation, if the efficacy of such approaches can be demonstrated they could be practically applied in the future with the advent of new technologies such as accurate pollution forecasting via machine learning models (Jumin et al., 2020; Wang et al., 2020). A holistic approach that considers the trade-offs between other abiotic stressors such as heat stress is needed, as irrigation plays a significant role in mitigating such stress (Zaveri and Lobell, 2019) and higher temperatures are a precursor of higher O$_3$ levels with the chance that O$_3$ effects are erroneously attributed to heat stress (Tai et al., 2014).

### 3.3 Effect of climate change on O$_3$ sensitivity

Higher O$_3$-induced yield losses were modelled under future scenarios (Fig. 5). For HUW234, a statistically significant increase in yield loss of 7.9±5.56% and 3.1±5.08% was modelled for RCP4.5 and RCP8.5 respectively, compared to the recent past climate. Similarly, for RCP4.5 and RCP8.5, an increase of 8.0±5.71% and 3.0±4.87% yield loss was predicted for HD-3118, respectively. This suggests that the increase in O$_3$ impact due to future emissions/climate is larger than the year-to-year variability in O$_3$ impact for the RCP4.5 (but not RCP8.5 where [O$_3$] were lower, see below) scenario. The recent past      climate represents the lowest mean O$_3$, suggesting O$_3$, rather than other environmental conditions that might influence sensitivity to O$_3$, (i.e. via alterations to stomatal O3 uptake)      is the most important factor in determining O$_3$-induced yield loss. These findings imply that the changing climate (i.e., higher frequency of temperatures that exceed the T$_{opt}$ with consequent reductions in stomatal conductance and hence O$_3$ flux) will be insufficient at ameliorating the increase in O$_3$-induced yield loss. This contrasts with several studies that have shown the potential of elevated temperatures to lead to reductions in O$_3$ flux via reduced stomatal conductance, thus reducing O$_3$ damage (Emberson et al., 2018; Feng et al., 2008). This could be due to differences in the timing and duration of periods of more extreme temperatures that exist between studies; a possibility that would benefit

from further study. It is important to clarify that in this study we explore future changes in ozone concentration due to changes
in climate due to changes in $O_3$ precursor emissions. $O_3$ studies often explore the effect of a 'climate change penalty' which is
the impact of a future climate on $O_3$ levels if emissions are held constant (Wu et al., 2008; Zanis et al., 2022). Although this is
not specifically investigated in this study it is worth noting that India is one of the global regions with the strongest effect of a
'climate change penalty' (Zanis et al 2022). Given the interplay between climate and $O_3$ in determining the extent of stomatal
$O_3$ uptake, and hence crop sensitivity to $O_3$, it is worth noting that even without changes in emissions, $O_3$-induced crop damage
would still be likely to change to some extent under future conditions.

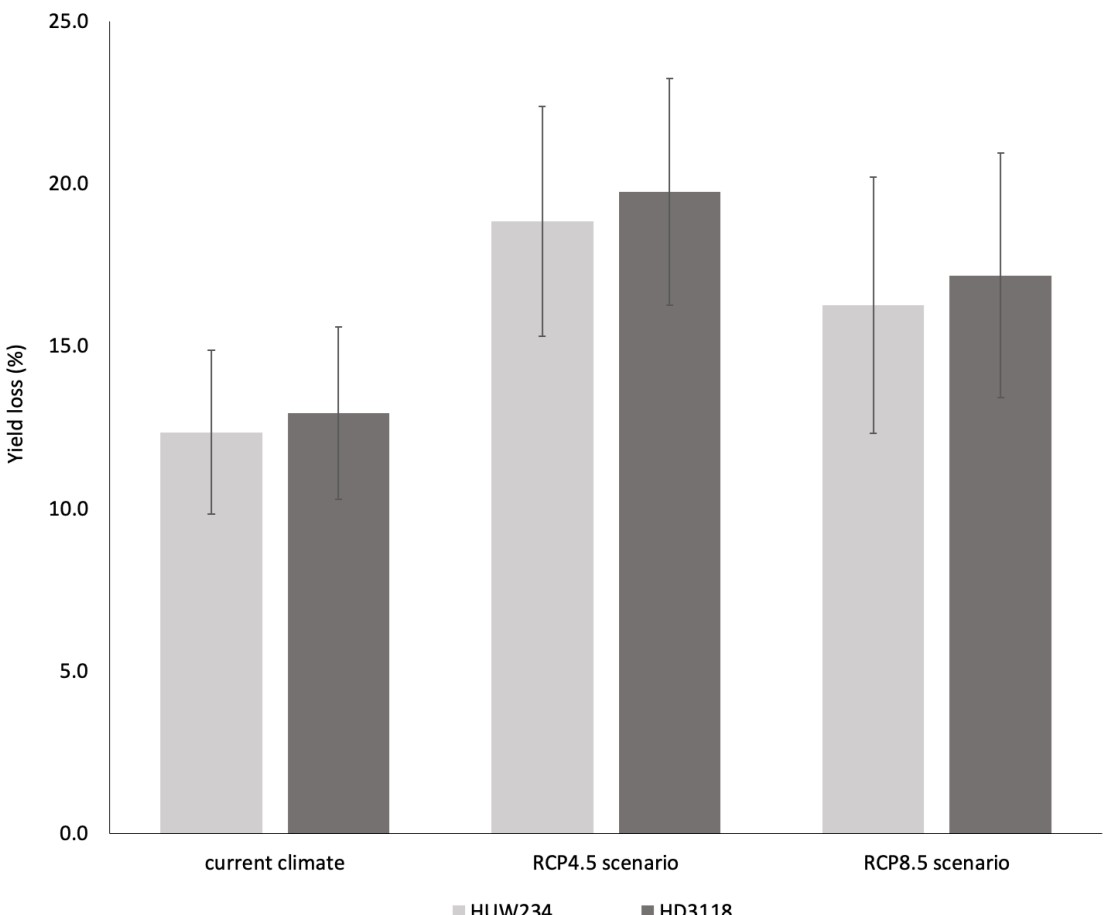


**Fig. 5: Mean $O_3$-induced relative yield losses (O3-RYL) ±SD modelled for the recent past     climate (1996-2005), and two future**
**climate scenarios for 2046-2055; RCP4.5 and RCP8.5 for two Indian spring wheat cultivars; HUW-234 and HD-3118.**
The results from this study are within the range published by Mills et al. (2018b) for $O_3$-induced yield losses for wheat in Uttar
Pradesh, which were modelled for 2010-2012 using $POD_3IAM$ with European wheat parameterisation and a broadscale
assessment of India's wheat growing season. The Mills et al. (2018b) study used the most recent methodology from CLRTAP
(2017) to calculate $O_3$-induced yield loss for wheat, as a reference POD$_3$IAM value representing $O_3$ uptake at pre-industrial
conditions was subtracted before crop loss was calculated. Whilst this study also uses a stomatal flux-based metric, POD$_3$IAM
is vegetation-type specific suited for large-scale modelling (CLRTAP, 2017). The POD$_6$SPEC was used in this current study
since we were able to define a cultivar-specific growth period with some certainty thereby allowing greater confidence in the
use of the more biologically relevant metric than POD$_3$IAM (CLRTAP, 2017). It is also worth noting that the timing of $O_3$ and
water stress may be important in predicting how plants respond to these stresses since $O_3$ has been found to damage stomatal
functioning causing plants lose the ability to respond to water stress (e.g. Wilkinson et al., 2010). Ideally, $O_3$ impact models
would include mechanisms that simulate $O_3$-induced loss in stomatal functioning, however, to our knowledge such modelling
mechanisms have not yet been developed or included and would likely require experimental data to identify thresholds at
which stomatal functioning is impaired.
Mean relative $O_3$ induced yield losses for both cultivars modelled under RCP4.5 (18.7±3.83%) were significantly higher than
the RCP8.5 (13.8±3.22%). This is likely due to a combination of slightly higher [$O_3$] in RCP4.5 (Table 1) and the WRF-Chem
model projections of higher temperatures (limiting $O_3$ flux as temperatures have a tendency to exceed T$_{opt}$) under RCP8.5.
Whilst the RCP4.5 scenario sees a global reduction in [$O_3$] due to pollution regulation, the South Asian region is an exception
to this rule, where [$O_3$] continues to increase at a similar rate as occurred in previous decades (Tai and Martin, 2017). RCP8.5
projects a worldwide increase in [$O_3$] due to the lack of regulation of precursor emissions except in parts of the US, East and
Southeast Asia (Tai and Martin, 2017). Therefore, mean [$O_3$] during the growing season is lowest in the recent past     climate
at 48.6ppb but similar, at least in South Asia, in both the RCP4.5 and RCP8.5 scenarios (60.5ppb and 59.7ppb respectively;
Table 1).
Relatively small differences in 2000-to-2050 increases in $O_3$ over South Asia between RCP4.5 and RCP8.5 have also been
found before (Tai et al., 2014; their Supplementary Figure 1). Our WRF-Chem model results do show a slightly higher increase
in $O_3$ precursors over India in RCP4.5 than RCP8.5 (not shown), likely explaining the slightly higher $O_3$ increase in the RCP4.5
scenario. However, several factors influence the modelled future $O_3$ concentration changes, such as the future change in
meteorological variables, the non-linearities of $O_3$ chemistry, and natural interannual variability. For example, Sharma et al.
(2023) found underestimation of relative humidity in meteorological data used in $O_3$ simulations which will have some
influence on $O_3$ production estimates. Future studies should utilize emission scenarios that are more updated in terms of air
pollution policies.
The $O_3$-induced yield loss will increase from current levels, regardless of whether global emissions follow a business-as-usual
or medium stabilisation scenario. We predict that $O_3$-induced yield losses will continue to increase in South Asia with climate
change, given the co-emission of radiative forcers and $O_3$ precursors and the two-way causality that exists between $O_3$
formation and climate change, i.e., hot, sunny conditions likely to be enhanced under climate change encourages $O_3$ formation,
whilst $O_3$ itself is a radiative forcer (Fu and Tian, 2019). This means that $O_3$ and climate variable stress are likely to co-occur
in the future which becomes especially problematic for crop productivity when environmental thresholds (e.g. due to
temperature extremes) for plant productivity are exceeded. South Asia and the IGP are important agricultural regions where
O$_3$ thresholds are being exceeded now (Mills et al., 2018c), with the likelihood that the extent of such exceedance will only
worsen in the future and with climate change (Cooper et al., 2014; Fowler et al., 2008; Fu and Tian, 2019; Rathore et al., 2023).
It should be noted that there are uncertainties in the WRF-Chem model used in modelling meteorological and [O$_3$] data. There
are important criticisms that the WRF-Chem model is limited in its ability in capturing true wind speeds, which influences
temperature and O$_3$ mixing ratios (Rydsaa et al., 2016). Despite this, these findings serve as a useful insight into the future risk
of O$_3$ on wheat yields relative to early 21$^{st}$-century conditions. Ideally, future research should consider the use of model
ensembles to more robustly capture ranges in future meteorological and [O$_3$] data.
**3.4 Influence of cultivar physiology on O$_3$ sensitivity**
The O$_3$.RYL modelled for HUW-234 were similar to HD-3118 in the recent past     climate and both the RCP4.5 and RCP8.5
scenarios (Fig. 5). This is due to the similarity in $g_{max}$ values for HUW-234 and HD-3118 which were estimated, from empirical
data at 500mmol O$_3$ m$^{-2}$ PLA s$^{-1}$ and 520.9 mmol O$_3$ m$^{-2}$ PLA s$^{-1}$ respectively. The mean yield losses for recent past     climate
and RCP scenarios combined, modelled for HD-3118 (14.5±0.05%) were similar to HUW-234 (14.3±0.05%), a difference of
0.2%. Despite the similar mean yield losses observed for HD-3118 and HUW-234, these results align with concerns that
modern wheat cultivars are more susceptible to O$_3$-damage as they are bred for maximum gas exchange or heat tolerance rather
than O$_3$ tolerance (Emberson et al., 2018; Pleijel et al., 2006; Yadav et al., 2020). Typically, plant traits bred for heat tolerance
and maximum gas exchange conflict with traits for O$_3$ tolerance and may increase irrigation requirements; i.e. higher stomatal
conductance enhances transpiration rates, allowing for higher rates of photosynthesis (Pleijel et al., 2007; Yadav et al., 2020).
Despite the potential for HD-3118 to produce higher yields due to a high stomatal conductance, HUW-234 performs better in
terms of O$_3$ tolerance for Varanasi's recent past     conditions and projections for the future climate and [O$_3$]. This is also
observed in the empirical data for Varanasi from 2016-18, which was used to parameterisation the ETS model. The empirical
data observed lower relative yield loss under elevated [O$_3$] compared to ambient [O$_3$] for HUW-234 than HD-3118 (21.2%
and 23.2% respectively; see Table S1). Absolute yields failed to observe the higher yielding potential expected of HD-3118
even under ambient conditions; the mean absolute grain yield for HUW-234 under ambient [O$_3$] was 533.4g/m compared to
432.8g/m for HD-3118. Under elevated [O$_3$], the yield gap widens; HUW-234 has an absolute grain yield of 420.4g/m whilst
HD-3118 has a yield of 332.3g/m. This suggests O$_3$ has a greater impact on yield in HD-3118 than HUW-234, possibly even
under ambient concentrations.
Despite corroborating literature for the DO$_3$SE model results (Yadav et al., 2020), there is some limitation in the ability to
accurately parameterise the model for specific cultivars. Here we have been able to parameterise key parameters that will
influence stomatal O$_3$ flux (g$_{max}$ and f$_{phen}$) for Indian varieties, however, the remaining parameters that determine the
modification of g$_{max}$ by environmental conditions rely on European parameterisations. Similarly, the DO$_3$SE model estimates
O$_3$-induced crop yield losses based on a dose-response relationship configured using five European wheat cultivars (CLRTAP,
2017). Whilst the DO$_3$SE model is a valuable tool for risk assessment, the use of appropriately calibrated and evaluated crop
models will provide mechanisms to fully explore the interplay between stresses such as O$_3$ and water stress on yield (Emberson

et al., 2018). The results of this paper make clear the need for such modelling to improve our understanding of how these different stresses act over the course of the growing season to determine changes in productivity. A new generation of crop models that are being developed to incorporate the $O_3$ effect, as well as other stresses (Emberson et al., 2018), will be able to explore trade-offs between stresses related to soil water, extreme temperatures, and soil fertility. Such advances in crop modelling will be crucial in assessing future wheat productivity under a range of abiotic stress conditions.

In this Indian study, the mid-anthesis and grain filling period occurred in March (Fig.'s 1 and 2) which corresponds to peak $O_3$ in Uttar Pradesh (Jain et al., 2023; Mukherjee et al., 2019; Shukla et al., 2017). However, a study on timely-sown Chinese winter wheat cultivars found that elevated $O_3$ only had a significant effect during the mid-grain filling stage, suggesting that timing mid-grain filling with $O_3$ troughs could be a mitigation strategy, which may be achieved by earlier sowing (Feng et al., 2016). Late planting results in reduced productivity of the wheat crop, with earlier, timely sowing of wheat in the third week of November yielding the best productivity in Eastern Uttar Pradesh (Chandna et al., 2004). Past studies have reported that delays in sowing after mid-November leads to reduction in yield of wheat, often at a rate of 1-1.5% per day (McDonald et al., 2022; Ortiz-Monasterio R. et al., 1994). In addition, Kumar et al. (2014) claimed conversion from late to timely-sown would offset the impacts of climate change. A multi-tolerance approach like early sowing could mitigate heat and $O_3$ stress however, late sowing is often due to delays in harvesting rice in Rice-Wheat systems, a cropping sequence which provides income for tens of millions of farm families (Jain et al., 2017; Mishra et al., 2021). Further investigation of the inter-play between [$O_3$] profiles over the growing season and targeted crop phenology of different cultivar types should be conducted.

**4 Conclusion**

Whilst irrigation has played a pivotal role in increasing wheat production in India through maximising yields, $O_3$ is likely to negate some of the yield benefits of irrigation, which will reduce irrigation efficiency. Based on the $POD_6SPEC$ values obtained *via* the $DO_3SE$ model and associated flux-response relationships, $O_3$ concentrations prevalent in the IGP region of India are high enough to cause grain yield losses in Indian wheat. This paper demonstrates the complexity of avoiding $O_3$-stress and the importance of taking a multi-stress approach to mitigation. Since high levels of $O_3$ typically coincide with other abiotic stressors such as heat stress, the approach taken to maximise crop yield must consider multiple stressors and their interactions. Rather than altering irrigation patterns to mitigate $O_3$ stress and risk increasing the effect of other stressors such as water stress or heat stress, earlier sowing to avoid peak $O_3$ and temperatures in March may benefit irrigated wheat growing in India. Given that modern wheat cultivars are more $O_3$-sensitive, wheat growers should reconsider using modern cultivars bred for optimal gas exchange.

**Competing interests**

The contact author has declared that none of the authors has any competing interests

**Author Contribution**

GE and LE designed the modelling study and GE and SB performed the $DO_3SE$ model runs. OH provided the modelled $O_3$ and meteorological data and advised on its use within the study. MA and DSY provided the experimental data used to calibrate the $DO_3SE$ model and advised on its use within the study. CON, CJ, JC and PP supported $DO_3SE$ model calibration and analysis of the results. GE prepared the manuscript with contributions from all co-authors.

**Acknowledgements**

Funding from The Norwegian Research Council funded CICERO strategic project (grant no. 160015/F40) and the CiXPAG project (grant no. 244551) provided support to Lisa Emberson, Øivind Hodnebrog and Madhoolika Agrawal. The Viking cluster was used during this project, which is a high-performance computing facility provided by the University of York. We are grateful for computational support from the University of York, IT Services and the Research IT team.

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
