# Peer review of "Ozone pollution may limit the benefits of irrigation to wheat"

_EGUsphere, 2024_

## Community Comment (CC1)

January 19, 2025

Comments by Owen R. Cooper (TOAR Scientific Coordinator of the Community Special Issue) on:

Ozone pollution may limit the benefits of irrigation to wheat productivity in India

Gabriella Everett, Øivind Hodnebrog, Madhoolika Agrawal, Durgesh Singh Yadav, Connie O'Neill, Chubamenla Jamir, Jo Cook, Pritha Pande, and Lisa Emberson

EGUsphere [preprint], https://doi.org/10.5194/egusphere-2024-3371
Discussion started Nov. 15, 2024
Discussion closes Jan. 24, 2025

This review is by Owen Cooper, TOAR Scientific Coordinator of the TOAR-II Community Special Issue. I, or a member of the TOAR-II Steering Committee, will post comments on all papers submitted to the TOAR-II Community Special Issue, which is an inter-journal special issue accommodating submissions to six Copernicus journals:  ACP (lead journal), AMT, GMD, ESSD, ASCMO and BG. The primary purpose of these reviews is to identify any discrepancies across the TOAR-II submissions, and to allow the author teams time to address the discrepancies.  Additional comments may be included with the reviews. While O. Cooper and members of the TOAR Steering Committee may post open comments on papers submitted to the TOAR-II Community Special Issue, they are not involved with the decision to accept or reject a paper for publication, which is entirely handled by the journal's editorial team.

**Comments regarding TOAR-II guidelines:**

TOAR-II has produced two guidance documents to help authors develop their manuscripts so that results can be consistently compared across the wide range of studies that will be written for the TOAR-II Community Special Issue.  Both guidance documents can be found on the TOAR-II webpage: https://igacproject.org/activities/TOAR/TOAR-II

*The TOAR-II Community Special Issue Guidelines*:   In the spirit of collaboration and to allow TOAR-II findings to be directly comparable across publications, the TOAR-II Steering Committee has issued this set of guidelines regarding style, units, plotting scales, regional and tropospheric column comparisons, and tropopause definitions.

*The TOAR-II Recommendations for Statistical Analyses*:  The aim of this guidance note is to provide recommendations on best statistical practices and to ensure consistent communication of statistical analysis and associated uncertainty across TOAR publications. The scope includes approaches for reporting trends, a discussion of strengths and weaknesses of commonly used techniques, and calibrated language for the communication of uncertainty. Table 3 of the TOAR-II statistical guidelines provides calibrated language for describing trends and uncertainty, similar to the approach of IPCC, which allows trends to be discussed without having to use the problematic expression, "statistically significant".

**General comments:**

Please provide some justification for selecting the period 1996-2005 as being representative of the current climate. The mid-point of this 10-year time-slice is 2000, and it is now 2025, so we are 2.5 decades past this period. The latest estimate from NASA (http://www.columbia.edu/~mhs119/Temperature) is that Earth's surface temperature is increasing at the rate of 0.18 C per decade. Therefore, the planet has warmed by 0.45 C since the year 2000. Given the clear warming of the Earth since the year 2000, the period 1996-2005 does not seem to be representative of the current climate. The CEDS emissions inventory has been updated to include recent years (Hoesly et al., 2024), so a more current period could be used.

Regarding Section 3.3 (Effect of climate change on O3 sensitivity), please make clear the type of climate impact(s) that you are investigating. Several studies have explored the impact of the "climate change penalty", which is the impact of a future climate on ozone levels if emissions are held constant (Wu et al., 2008; Zanis et al., 2022). Your analysis does not seem to be addressing the "climate change penalty", and to avoid any confusion, please make this point clear. But please consider the impacts of the climate change penalty on your study region, as Zanis et al. (2022) concluded that India and China would be the regions with the strongest impacts.

**References**

Hoesly, R., Smith, S. J., Prime, N., Ahsan, H., Suchyta, H., O'Rourke, P., Crippa, M., Klimont, Z., Guizzardi, D., Behrendt, J., Feng, L., Harkins, C., McDonald, B., Mott, A., McDuffie, A., Nicholson, M., & Wang, S. (2024). CEDS v_2024_07_08 Release Emission Data (v_2024_07_08) [Data set]. Zenodo. https://doi.org/10.5281/zenodo.12803197

Wu, S., Mickley, L. J., Leibensperger, E. M., Jacob, D. J., Rind, D., and Streets, D. G.: Effects of 2000–2050 global change on ozone air quality in the United States, Journal of Geophysical Research: Atmospheres, 113, D06 302, https://doi.org/10.1029/2007JD008917, 2008.

Zanis, P., Akritidis, D., Turnock, S., Naik, V., Szopa, S., Georgoulias, A.K., Bauer, S.E., Deushi, M., Horowitz, L.W., Keeble, J. and Le Sager, P., 2022. Climate change penalty and benefit on surface ozone: a global perspective based on CMIP6 earth system models. Environmental Research Letters, 17(2), p.024014.

---

## Author Comment (AC1)

**Editor**

We thank the editor for their valuable comments which have greatly improved our manuscript. We address the specifics of their feedback below.

**General comments:**

Please provide some justification for selecting the period 1996-2005 as being representative of the current climate. The mid-point of this 10-year time-slice is 2000, and it is now 2025, so we are 2.5 decades past this period. The latest estimate from NASA (http://www.columbia.edu/~mhs119/Temperature) is that Earth's surface temperature is increasing at the rate of 0.18 C per decade. Therefore, the planet has warmed by 0.45 C since the year 2000. Given the clear warming of the Earth since the year 2000, the period 1996-2005 does not seem to be representative of the current climate. The CEDS emissions inventory has been updated to include recent years (Hoesly et al., 2024), so a more current period could be used.

We acknowledge this point and agree with the reviewer that we cannot claim 1996- 2005 is a current climate. Since the point of the paper is to show how climate change changes in ozone concentration might influence the relative importance of both combined and individual water and ozone stress we change 'current' to 'recent past' making clear that we select time periods to show substantial differences in both climate and ozone levels.

Regarding Section 3.3 (Effect of climate change on O3 sensitivity), please make clear the type of climate impact(s) that you are investigating. Several studies have explored the impact of the "climate change penalty", which is the impact of a future climate on ozone levels if emissions are held constant (Wu et al., 2008; Zanis et al., 2022). Your analysis does not seem to be addressing the "climate change penalty", and to avoid any confusion, please make this point clear. But please consider the impacts of the climate change penalty on your study region, as Zanis et al. (2022) concluded that India and China would be the regions with the strongest impacts.

This has now been made clear in Section 3.3 by adding the following text to the end of this section:-

*'It is important to clarify that in this study we explore future changes in ozone concentration due to changes in climate due to changes in O3 precursor emissions. Ozone studies often explore the effect of a 'climate change penalty' which is the impact of a future climate on O3 levels if emissions are held constant (Wu et al., 2008; Zanis et al., 2022). Although this is not specifically investigated in this study it is worth noting that India is one of the global regions with the strongest effect of a 'climate change penalty' (Zanis et al 2022). Given the interplay between climate and O3 in determining the extent of stomatal O3 uptake, and hence crop sensitivity to O3, it is worth noting that even without changes in emissions, O3-induced crop damage would still be likely to change to some extent under future conditions'.*

References
Hoesly, R., Smith, S. J., Prime, N., Ahsan, H., Suchyta, H., O'Rourke, P., Crippa, M., Klimont, Z., Guizzardi, D., Behrendt, J., Feng, L., Harkins, C., McDonald, B., Mott, A., McDuffie, A., Nicholson, M., & Wang, S. (2024). CEDS v_2024_07_08 Release

Emission Data (v_2024_07_08) [Data set]. Zenodo.
https://doi.org/10.5281/zenodo.12803197

Wu, S., Mickley, L. J., Leibensperger, E. M., Jacob, D. J., Rind, D., and Streets, D. G.: Effects of 2000–2050 global change on ozone air quality in the United States, Journal of Geophysical Research: Atmospheres, 113, D06 302, https://doi.org/10.1029/2007JD008917, 2008.

Zanis, P., Akritidis, D., Turnock, S., Naik, V., Szopa, S., Georgoulias, A.K., Bauer, S.E., Deushi, M., Horowitz, L.W., Keeble, J. and Le Sager, P., 2022. Climate change penalty and benefit on surface ozone: a global perspective based on CMIP6 earth system models. Environmental Research Letters, 17(2), p.024014

---

## Author Comment (AC2)

**Reviewer 1**

**General comment:**

This paper presents ozone and water-stress impact on wheat productivity under current and future climate scenarios in India to highlight the importance of the irrigation for crop production. The effects of ozone stress on the yield under the different humidity and climate conditions were quantified based on WRF model simulation, and their sensitivities influenced by climate and plant were discussed. Overall, the study addresses a significant gap in the understanding of the agricultural impact of ozone and its coupling with water stress in India, a region experiencing increasing ozone pollution. The conception is interesting and impressive. However, authors only discuss the average result of ozone impact during the study period, the spatial and temporal variations of ozone impact are not included, which need more comprehensive investigations. Additionally, the method introduction should be more specific and detailed.

The following changes should be made to improve the manuscript with respect to clarity, information content and thus value to the reader.

We thank the reviewer for their insightful comments which have greatly improved the quality of the manuscript, we address the specifics of their feedback below.

**Specific questions/issues:**

Table 1 : Temperature(°c)→℃
Corrected

Line 192: How to calculate the RstoH2O? based on Gsto?
Yes, it is the inverse of stomatal conductance to H2O, we add this into the manuscript to provide extra clarity to the reader.

Line 199-200: "K$y$ is the crop-specific yield response factor assumed to be 1.15 for the whole growing season" Are there any difference for the response factor of different wheat cultivars? as the different stomatal sensitivity and gmax.
The K$y$ values come from the FAO crop response to irrigation and drainage report, where 1.15 is the standard value reported for spring wheat. While it is highly likely that this value will vary slightly between different cultivars due to multiple cultivar-specific factors including gas exchange and photosynthesis, 1.15 is considered the average for spring wheat, while for winter wheat it is 1.05. In the absence of a specific value for Indian wheat, but knowing spring wheat cultivars are grown in the country, we use the spring wheat value.

Table 2: Canopy height: How about the variability of canopy height? Canopy height depends on wheat growth, and it increases from 0 m after sowing to the maximum height (~ 1 m) at the end of growth stage. Why do you choose the maximum height as canopy height in the simulation? How does the height affect the simulation? (Crop height: https://doi.org/10.1007/s12524-024-02028-4)
The canopy height affects how the O3 concentrations are propagated from the height at which they were measured to the crop canopy level. The accumulation of stomatal O3 flux using the method described in the present study begins at approximately midanthesis (since mid anthesis to maturity is the O3 sensitive period for wheat). At mid-anthesis the crop can be assumed to be at its maximum height hence the choice of maximum height for the simulation.

Section 2: The method of chemistry simulation and evaluation in WRF model should be added, as ozone concentration is critical to ozone uptake and impact.
We have now made clear that the WRF-Chem model description provided in section 2.2 is for both meteorological and $O_3$ concentration data, the references provided give additional information on the $O_3$ chemistry used in the WRF-Chem model. A paper by Sharma et al. (2017) describing evaluation of the WRF-Chem model is included in this section:

"*WRF-Chem with the RADM2 chemical mechanism has been found to reproduce diurnal average ozone over India in February-May relatively well, while noontime ozone concentrations show considerable differences between simulations with various emission inventories (Sharma et al., 2017)*."

Section 3.1: Although phenology is crucial for leaf stomatal conductance and ozone uptake, the magnitude of ozone uptake is the most important for the accurate evaluation of ozone impact on yield loss. How do you assess the uncertainty of modelled POD?
We thank the reviewer for spotting that we have not included details of the certainty with which the DO3SE model can estimate PODy. We have now added text in Section 3.1 (which has been renamed to 'Phenology and stomatal ozone uptake') to provide additional details of how well the DO3SE model performs in simulating stomatal ozone flux.

Section 3.2: What is the definition of water-stress condition? Are there any indexes or indicators to measure the humidity? As POD is a cumulative metric for the whole period of wheat growth, how do you simulate the cumulative ozone flux under two humidity conditions? How about the spatial distribution of O3-related yield loss and WSRYL?
Water stress is defined within the text at line 112 as 'yield losses due to water stress [being] based on yield responses to the ratio of actual vs potential evapotranspiration (cf. FAO (2012))'. Soil humidity is defined by plant available water (PAW) which is defined in Table 2 and associated text, we now include units for PAW in the notes to the Table. PAW is also a cumulative variable (increasing and decreasing depending upon the balance between incoming precipitation and outgoing evapotranspiration). Since the estimate of PAW is intrinsically linked to $g_{sto}$ via $R_{stoH2O}$ (see Eq 5) the estimates of different levels of soil water stress are internally consistent with estimates of ozone flux. The spatial distribution of $O_3$- and water-stress-relative yield loss is not considered in this study as the model has been applied at a site-level.

Line 289-292: "Under rainfed conditions, mean O3RYL was projected to be negligible (0.6%), significantly lower than the mean O3RYL when no water-stress is assumed under irrigation (10.7% with a range of 4.8-15.4%). This demonstrates the importance of irrigation for wheat production in India and highlights the substantial influence on the yield of O3 for irrigated wheat." What are the main pathways through which the irrigation could reduce ozone impact? and their relative contributions?

Irrigation will actually increase ozone impact since irrigated conditions will increase stomatal conductance since there is no need for the stomates to close to conserve

water. Irrigation will also increase transpiration which will reduce the leaf to air vapour pressure deficit which will also lead to an increase in stomatal conductance. These mechanisms are included in the introduction (i.e. see line 79) therefore no additional action is taken to address this comment.

Lines 328-330: "The current climate represents the lowest mean O3, suggesting O3, rather than other environmental conditions that might influence sensitivity to O3, is the most important factor in determining O3-induced yield loss." Please rephrase this sentence, it's a bit hard to understand. Whose sensitivity do the environmental conditions influence?

Sentence modified to '*The recent past current climate represents the lowest mean $O_3$, suggesting $O_3$, rather than other environmental conditions that might influence sensitivity to $O_3$, (i.e. via alterations to stomatal $O_3$ uptake) is the most important factor in determining $O_3$-induced yield loss*' to make it easier to understand.

Lines 350-355: "Whilst the RCP4.5 scenario sees a global reduction in [O3] due to pollution regulation, the South Asian region is an exception to this rule, where [O3] continues to increase at a similar rate as occurred in previous decades (Tai and Martin, 2017). RCP8.5 projects a worldwide increase in [O3] due to the lack of regulation of precursor emissions except in parts of the US, East and Southeast Asia (Tai and Martin, 2017). Therefore, mean [O3] during the growing season is lowest in the current climate at 48.6ppb but similar, at least in South Asia, in both the RCP4.5 and RCP8.5 scenarios (60.5ppb and 59.7ppb respectively; Table 1)"

Ozone concentration continues to increase at a similar rate under the RCP4.5 emission scenario, and RCP8.5 also will lead a worldwide increase to ozone due to the lack of emission regulation. So why are ozone concentrations at the similar levels under RCP4.5 and RCP 8.5? and ozone in the RCP4.5 is even slightly higher than that in the RCP8.5.

We agree that the similarity in future ozone concentration change between RCP8.5 and RCP4.5 may sound counterintuitive. We have looked into the WRF-Chem model results for some of the ozone precursors, NOx and hc3 (a lumped NMVOC compound covering several alkanes), and their future changes over India are also similar between the two scenarios, with somewhat larger increases for RCP4.5 than RCP8.5:

[Figure]

*10-year averaged WRF-Chem results of 2046-2055 minus 1996-2005 concentration changes for December-April zoomed in over India.*

These NOx and hc3 concentration changes are strongly influenced by changes in local NOx and NMVOC emissions, and are likely to explain why O₃ is slightly higher in RCP4.5 than RCP8.5, although several factors contribute. To shed some more light on this issue, we have added the following after the above-mentioned text:

"*Relatively small differences in 2000-to-2050 increases in O3 over South Asia between RCP4.5 and RCP8.5 have also been found before (Tai et al., 2014; their Supplementary Figure 1). Our WRF-Chem model results do show a slightly higher increase in O3 precursors over India in RCP4.5 than RCP8.5 (not shown), likely explaining the slightly higher ozone increase in the RCP4.5 scenario. However, several factors influence the modelled future ozone concentration changes, such as the future change in meteorological variables, the non-linearities of ozone chemistry, and natural interannual variability. Future studies should utilize emission scenarios that are more updated in terms of air pollution policies.*"

---

## Author Comment (AC3)

**Reviewer 2**

**General Comments:**

The paper describes model study to assess the inter-related impacts of ozone- and water-stress-induced yield losses in two wheat cultivars in an agricultural region in India. The authors model ozone fluxes into the plant using WRF-Chem simulations of current and future surface ozone and meteorological conditions, using the RCP4.5 and RCP8.5 future scenarios. The study finds that under rainfed conditions, water stress induces stomatal closure, thus protecting plants from ozone uptake and minimizing ozone-induced yield losses, though water-stress-induced losses remain. However, under irrigated conditions, stomatal opening sustains high ozone uptake, resulting in substantial ozone-induced yield losses that largely offset reductions in water-stress-induced losses. The authors advocate for more research into ozone-water effects in crops and suggest potential changes in irrigation management to find practicable strategies for mitigating wheat yield losses due to both water stress and ozone.

We thank the reviewer for their insightful feedback which has enabled us to substantially improve the quality of our manuscript. We address the specifics of their feedback below.

**Specific Comments:**

Section 2.3-2.4: I have seen some studies (not treating wheat) that suggest the timing of ozone exposure and water stress may be important in predicting how they interact to impact plant damages (e.g., Matyssek 2006; https://doi.org/10.1055/s-2005-873025). For example, if ozone exposure preceding drought damages stomatal regulation, this could impact stomatal response to VPD under water stress. Can you please expand on whether the DO$_3$SE model has any mechanism to test this type of asynchrony in drought or ozone exposure leading to injury or impairment of stomatal regulation? Could this be applied in such a study?

Unfortunately, the current version of the DO$_3$SE model is unable to simulate O$_3$ impact on stomatal functioning. However, the reviewer raises an important point here. To address this we now include an additional few sentences in section 3.3.

'*The timing of O$_3$ and water stress may be important in predicting how plants respond to these stresses since O$_3$ has been found to damage stomatal functioning causing plants lose the ability to respond to water stress (e.g. Wilkinson et al., 2010). Ideally, O$_3$ impact models would include mechanisms that simulate O$_3$-induced loss in stomatal functioning, however, to our knowledge such modelling mechanisms have not yet been developed or included and would likely require experimental data to identify thresholds at which stomatal functioning is impaired*'.

Section 2.3: Are there any data to validate the DO3SE derived ozone fluxes, even at other field sites? What is the uncertainty in model derived fluxes?

The wheat stomatal ozone flux model used in the DO$_3$SE model has been evaluated against wheat stomatal conductance data (the primary determinant of stomatal ozone flux) collected under experimental conditions in Ostad, Sweden (Pleijel et al 2007) and found to perform well (with an R2 value of 0.83 for regression of observed against

modelling $g_{sto}$. The $DO_3SE$ model has also been extensively evaluated at a number of crop sites (as reported in Tuovinen, Clifton, Emmerichs (sub)) although these evaluations rely on total ozone flux and deposition measurements (since they use ozone flux tower data) so incorporate whole canopy ozone deposition and flux rather than representative upper leaf stomatal ozone flux required for PODy calculations. The combination of these evaluation methods that focus on leaf level stomatal conductance and canopy level stomatal ozone flux model together provide confidence in the predictive abilities of the $DO_3SE$ model. To make this clearer in the manuscript we have incorporated the following sentence in the discussion at the end of a renamed section 3.1 (see also comment from Reviewer 1).

*'The $DO_3SE$ wheat stomatal ozone flux model has been evaluated against wheat gsto data (the primary determinant of stomatal ozone flux) collected under experimental conditions in Ostad, Sweden (Pleijel et al. 2007) and found to perform well (with an $R^2$ value of 0.83 for a regression of observed against modelled gsto). The $DO_3SE$ model has also been extensively evaluated for a number of crops at locations around the world (as reported in Tuovinen et al. (2004) for wheat growing near Cumono novo in Italy and Emmerichs et al. (2025) for wheat growing near Grignon in France). These evaluations rely on total ozone flux and deposition measurements (since they use ozone flux tower data) or water flux measurements and thereby test whole canopy fluxes rather than the representative upper leaf stomatal flux required for PODy calculations. The combination of these evaluation methods focussing on both leaf level stomatal conductance and canopy level ozone flux together provide confidence in the predictive abilities of the $DO_3SE$ model'.*

Equation 4: Some more clarification would be helpful. Where does the factor of 3.85, for example, come from? Can you provide a reference for this?
We have now added a sentence to the manuscript to explain where this regression comes from along with the corresponding citations.

Line 273-76/Figure 3: It is compelling to see a plot of WRF vs site temperatures over the growing season. The authors should expand a bit on how they expect modeled meteorology to impact their conclusions. Does WRF under-predict humidity for example? I think it would also be very compelling to see a similar plot of modeled and observed ozone at this site, since according to the authors, this site was chosen in part due to availability of ozone data (L116-18).

We have now included text describing an evaluation of the WFR-Chem model (in response also to a comment by Reviewer 1). We also refer to details of how modelled meteorology influences $O_3$ concentration modelling in relation to an underestimation of relative humidity as reported by Sharma et al. (2023).

We agree that it will be useful to add a plot of modelling and observed [$O_3$] data to complement the existing Fig 3 for surface air temperature and will include this in the revised paper.

**Technical Corrections:**

L26-7: O3-RYL is given a slightly different meaning later on (see L287)
We thank the reviewer for spotting this typo. The discussion should use the term 'ozone relative yield loss' as defined earlier in the text. We have now ensured that O3-RYL for any RYL related to O3 stress and WS-RYL for any related to water stress.

L109: POD-SEPC acronym not previously defined
We now define $POD_6SPEC$ prior to use

L189: Symbol of Et is elsewhere fully capitalized
The Et symbol in eq. 5 has been corrected to $ET_a$